# Hearing Healthcare Professionals' Views about Over-The-Counter (OTC) Hearing Aids: Analysis of Retrospective Survey Data

Vinaya Manchaiah [1,2,3,4,5,*,†], Anu Sharma [6], Hansapani Rodrigo [3,7], Abram Bailey [8], Karina C. De Sousa [3,4] and De Wet Swanepoel [1,3,4,9]

1   Department of Otolaryngology-Head and Neck Surgery, University of Colorado School of Medicine, Aurora, CO 80045, USA
2   UCHealth Hearing and Balance, University of Colorado Hospital, Aurora, CO 80045, USA
3   Virtual Hearing Lab, Collaborative Initiative between the University of Colorado and the University of Pretoria, Aurora, CO 80045, USA
4   Department of Speech-Language Pathology and Audiology, University of Pretoria, Pretoria 0001, South Africa
5   Department of Speech and Hearing, Manipal College of Health Professions, Manipal Academy of Higher Education, Manipal 576104, India
6   Brain and Behavior Laboratory, Department of Speech Language and Hearing Sciences, Institute of Cognitive Science, Center for Neuroscience, University of Colorado Boulder, Boulder, CO 80305, USA
7   School of Mathematical and Statistical Sciences, University of Texas Rio Grande Valley, Edinburg, TX 78539, USA
8   Hearing Tracker Inc., Austin, TX 40702, USA
9   Ear Science Institute Australia, Nedlands, WA 6009, Australia
*   Correspondence: vinaya.manchaiah@cuanschutz.edu; Tel.: +1-(303)-724-9316; Fax: +1-(303)-724-1961
†   Current address: School of Medicine, University of Colorado, Aurora, CO 80045, USA.

**Abstract:** Over-the-counter hearing aids have been available to consumers in the US since 17 October 2022 following a ruling by the Food and Drug Administration. However, their reception by hearing healthcare professionals (HHP) has been mixed, and concerns have been expressed by many HHPs. The aim of this study was to examine the concerns that HHPs have towards over-the-counter (OTC) hearing aids. The study used a retrospective survey design. The survey data of HHPs (n = 730) was obtained from Hearing Tracker. A 22-item structured questionnaire was administered using a Question Scout platform. Descriptive analyses examined reported areas of concern and a Fisher's exact test examined the relationship between demographics and responses. A cluster analysis with partitioning around medoids (PAM) was used to identify a sub-group of participants based on responses. Nearly half of HHPs who participated reported that they will support patients with OTC hearing aids purchased elsewhere, whereas a quarter reported that they will sell OTC hearing aids in their clinic or website. HHPs expressed over 70% agreement in 'concern' statements in 14 of the 17 items. Issues about safety, counseling, and audiological care were the key concerns expressed by HHPs about OTC hearing aids. Some demographics (i.e., profession, primary position) were associated with responses to some statements. Two groups were identified based on the responses to concern statements. The HHPs in the first cluster 'OTC averse' (51%) agreed on all the 17 concern statements, whereas the second cluster 'OTC apprehensive' (49%) had some items rated as disagree (i.e., consumers will give up on amplification) and neither agree nor disagree (i.e., do not provide good value, warranties and return periods will be worse), and remaining items were rated as agree. OTC hearing aids were initiated to improve affordability, accessibility, and hearing aid uptake and are currently a rapidly emerging category of hearing devices. Overall, the results of the current study indicate that HHPs have serious concerns about OTC hearing aids. HHP concerns cited in this study provide useful feedback to stakeholders (e.g., HHP professional agencies, FDA, industry, and insurance payers) involved in improving OTC hearing aid implementation.

**Keywords:** hearing aids; over-the-counter hearing aids; direct-to-consumer hearing devices; professional perspectives; hearing healthcare professionals; audiologists

## 1. Introduction

Hearing loss is one of the most frequent chronic conditions in older adults with over 430 million people across the globe requiring rehabilitation [1]. Hearing aids are the most common management options for individuals with hearing loss [2]. However, hearing aid uptake remains low with only one in four people in the U.S. obtaining hearing aids [3]. The reason for this low uptake is multifaceted, although some of the main reasons include low perceived need, stigma, and high cost of the device [4]. In order to address access and affordability issues, the U.S. Congress passed the Over-the-Counter (OTC) Hearing Aid Act in 2017 mandating that the U.S. Food and Drug Administration (FDA) create a new hearing aid category for OTC hearing aids [5]. The FDA's final ruling on 16 August 2022 allows OTC hearing aids to be sold directly to consumers without the consultation of hearing healthcare professionals (HHPs) from 17 October 2022 in the US [6].

OTC hearing aids add to a range of hearing devices in the U.S. market including prescription hearing aids and various direct-to-consumer hearing devices such as personal sound amplification products (PSAPs) and hearables [7,8]. The OTC hearing aid category supersedes the direct-to-consumer (DTC) hearing aid category which was previously available for purchase online [9]. Companies who have legally sold DTC hearing devices prior to 16 October 2022 have until 14 April 2023 to comply with the labeling and packaging requirements set by the new OTC Hearing Aid Regulations.

The availability of OTC hearing aids has already created new service delivery models which can be primarily grouped into three categories [8]. First, OTC hearing aids can be purchased online or in-store without consulting HHPs in which the users self-select, self-fit, and self-manage the devices. Second, several companies that sell OTC hearing aids also provide remote customer service or clinical services to users. Finally, some HHPs are starting to dispense OTC hearing aids in their clinics and/or providing consultation sessions to select, fit, and troubleshoot the devices as required by the user. In the latter model, it is critical that HHPs understand the benefits and limitations of OTC HAs as well as recognize who may be a good candidate for OTC hearing aids in order to ensure users obtain optimal benefits. For this reason, it is important to understand HHPs' current understanding and views towards OTC hearing aids.

Discussion and development of an OTC hearing aid category started over a decade ago following the recommendations from the National Academies of Sciences, Engineering, and Medicine (NASEM), President's Council of Advisors on Science and Technology (PCAST), and the 2009 National Institute on Deafness and Other Communication Disorders (NIDCD) working group [10]. HHPs and the hearing device industry as well as professional organizations initially expressed significant concerns that OTC hearing aids will do more harm than good [11]. More recently professional associations such as the American Academy of Audiology (AAA) and the American Speech-Language and Hearing Association (ASHA) have changed their messaging to be more neutral and positive. However, HHPs continue to express concerns about OTC hearing aids in various forums (e.g., social media posts and popular media interviews). Given that HHPs currently fit most users with prescription hearing aids, their view on newly available OTC hearing aids presents an opportunity to better understand their concerns regarding this device category and evaluate their perceptions systematically against the available evidence.

The primary aim of this study was to examine HHP views regarding OTC hearing aids. The secondary aims included understanding how HHP professional positions interacted with their views on OTC hearing aids and identifying sub-group of HHPs based on their views towards OTC hearing aids.

## 2. Method

### 2.1. Study Design

The study used a retrospective survey design using an anonymous survey of HHPs on their views towards OTC hearing aids. Institutional review board approval was obtained from the research ethics committee (Reference number 02606623-HUM007/0922) at the University of Pretoria, South Africa prior to data analysis. There was no informed consent as the survey data was not part of a research study, although participants were aware that the data can potentially be used for research purposes.

### 2.2. Data Collection

Data was collected by a cross-sectional online survey administered by Hearing-Tracker.com using the Question Scout platform using convenience sampling. The survey link was posted in several audiology Facebook groups and for those audiologists who are registered for audiologist informer newsletters at HearingTracker.com in August 2022, immediately after the FDA ruling was released. The users of this group were encouraged to complete the survey anonymously and no incentive was given to participants. No strict inclusion/exclusion criteria were used. However, all users of this community who were mainly HHPs were encouraged to complete the survey. The survey included a 22-item structured questionnaire of which 17 items were focused on examining the HHPs' concerns about OTC hearing aids and the remaining 5 questions gathered some basic demographics (i.e., profession, primary practice type, primary position, OTC hearing aid association). The questions on concerns about OTC hearing aids were rated on a 5-point Likert scale (strongly agree with a value of 5 to strongly disagree with a value of (1), with agreement suggesting a more negative view or concern about OTC hearing aids. These 17 items were grouped into 5 domains that included HPP's views regarding (a) safety, (b) concerns about device handling and self-adjustment, (c) concerns about service delivery model, (d) concerns about counseling and audiological care, and (e) concerns about the optimal benefit and adverse events. Seven hundred and thirty HHPs completed the survey.

### 2.3. Data Analyses

Analyses were performed using R software (version 4.2.2). Concerns towards OTC hearing aids for each of the 17 items expressed by HHPs were analyzed using descriptive statistics. Fisher's exact test was used to examine the relationship between demographic variables of HHPs (i.e., profession and position) and their views towards OTC hearing aids. The unadjusted odds ratio was also calculated and reported. A Fisher's exact test was performed on each of the 17 items with the two demographic variables, although only significant associations are reported in the manuscript. Finally, cluster analysis with partitioning around medoids (PAM) was used to identify a sub-group of HHPs based on their views towards OTC hearing aids.

## 3. Results

### 3.1. Participant Demographics

Table 1 presents the participants' demographics. HHPs mainly consisted of audiologists (n = 653; 89.5%), followed by business owners, hearing instrument specialist (HIS), and Doctorate of Audiology (AuD) students. The mean years in practice was 16.7 years (SD = 12.4 years). In terms of the primary position, practice employee (n = 359; 49.2%) was the largest group followed by owner and practice manager. A total of 55.5% of the HHPs reported willingness to support patients with OTC hearing aids purchased elsewhere, whereas only 26.7% reported that they will sell OTC hearing aids in their clinic or website.

**Table 1.** Demographic information (n = 730).

| Variable | N (%) |
|---|---|
| Profession | |
| ▪ Audiologist | 653 (89.4%) |
| ▪ Business owner | 54 (7.4%) |
| ▪ Hearing Instrument Specialist | 18 (2.5%) |
| ▪ Students (i.e., Doctorate of Audiology degree students) | 5 (0.7%) |
| Primary practice type | |
| ▪ Independent private practice | 314 (43%) |
| ▪ Ear, Nose, and Throat (ENT) practice | 140 (19.2%) |
| ▪ Hospital | 73 (10%) |
| ▪ Manufacturer owned practice | 28 (3.8%) |
| ▪ University clinic | 25 (3.5%) |
| ▪ Corporate-owned practice | 33 (4.5%) |
| ▪ Veterans Administration | 27 (3.7%) |
| ▪ Big box store (e.g., Costco, Sam's club) | 5 (0.7%) |
| ▪ Group practice | 22 (3%) |
| ▪ Independent chain | 21 (2.9%) |
| ▪ Other | 42 (5.7%) |
| Primary position | |
| ▪ Other | 41 (5.6%) |
| ▪ Practice employee | 359 (49.2%) |
| ▪ Owner | 231 (31.6%) |
| ▪ Practice manager | 99 (13.6%) |
| OTC hearing aid association (select all that apply) | |
| ▪ I currently dispense hearing aids | |
| ▪ I will support patients with over-the-counter (OTC) hearing aids purchased elsewhere | 622 (85.2%) |
| | 405 (55.5%) |
| ▪ I will sell OTC hearing aids in my clinic or on my website | 195 (26.7%) |
| ▪ I will offer unbundled prices to compete with OTC hearing aid price | 309 (42.3%) |
| | 638 (87.3%) |
| ▪ I employ best practices or plan to improve my standard. | |

### 3.2. HHPs Concerns towards OTC Hearing Aids

Table 2 presents HHPs' concerns towards OTC hearing aids in terms of percentage response to each of the 17 items on a 5-point scale, as well as mean values. The mean responses for the 17 items ranged from 3.4 to items in the domain of concerns about the optimal benefit and adverse events (i.e., *do not provide good value* and *consumers will give up amplification*) to 4.5 to a few items in the domain of concerns about device handling and self-management (i.e., *consumers will struggle to identify and address common problems*) and the domain on concerns about counseling and audiological care (i.e., *best practice audiological care is more important*). Figure 1 presents the agreement (i.e., strongly agree and agree) or disagreement (i.e., disagree, strongly disagree) about concerns by combining the positive and negative items together. The proportion (%) of participants agreeing with the items of concern ranged from 44% for the item *do not provide good value* to 93.4% for the item on *will not be educated on effective communication strategies*. Moreover, for 14 of the 17 items, HHPs expressed over 70% agreement to concern statements. Issues about safety as well as about counseling and audiological care were the key concerns expressed by HHPs about OTC hearing aids. Overall, these results suggest that HHPs have serious concerns about OTC hearing aids.

**Table 2.** Hearing healthcare professionals (HHPs) concerns towards over-the-counter (OTC) hearing aids.

| Questions | % of Respondents | | | | | Mean (SD) | 95% CI (Lower to Upper) |
|---|---|---|---|---|---|---|---|
| | **Strongly Agree** | **Agree** | **Neither Agree nor Disagree** | **Disagree** | **Strongly Disagreed** | | |
| **Concerns About Safety** | | | | | | | |
| Not able to accurately predict hearing loss | 49.3 | 40 | 4.8 | 4.7 | 1.2 | 4.3 (0.9) | (4.3–4.4) |
| Miss medical red flags | 55.8 | 33.8 | 5.1 | 4.9 | 0.4 | 4.4 (0.8) | (4.3–4.5) |
| Significant safety risk | 31.1 | 44.8 | 10.3 | 11.9 | 1.9 | 3.9 (1.0) | (3.8–3.9) |
| **Concerns about device handling and self-adjustment** | | | | | | | |
| Struggle to insert hearing aids | 23.8 | 48.8 | 17.4 | 8.8 | 1.2 | 3.9 (0.9) | (3.8–3.9) |
| Consumers will struggle to program | 34.5 | 42.2 | 15.8 | 6.4 | 1.1 | 4.0 (0.9) | (3.9–4.1) |
| Consumers will struggle to identify and address common problems | 56 | 36.6 | 4.4 | 2.3 | 0.7 | 4.5 (0.8) | (4.4–4.5) |
| **Concerns about service delivery model** | | | | | | | |
| Lead to greater confusion | 38.9 | 41.4 | 10.4 | 7.5 | 1.8 | 4.1 (0.9) | (4.0–4.2) |
| Ripped off by bad actors | 35.9 | 35.7 | 17.8 | 8.8 | 1.8 | 3.9 (1.0) | (3.8–3.9) |
| Warranties and return periods will be worse | 24.2 | 26 | 38.1 | 9.9 | 1.8 | 3.6 (1.0) | (3.5–3.7) |
| **Concerns about counseling and audiological care** | | | | | | | |
| Will not be educated on hearing protection | 36.8 | 43.8 | 13.7 | 4.7 | 1 | 4.1 (0.9) | (4.0–4.2) |
| Will not be educated on realistic expectations | 54 | 38.3 | 4.5 | 2.5 | 0.7 | 4.4 (0.8) | (4.4–4.5) |
| Will not be educated on effective communication strategies | 51 | 42.4 | 4 | 2.1 | 0.5 | 4.4 (0.7) | (4.4–4.5) |
| Best practice audiological care is more important | 67 | 24.8 | 4.7 | 2.7 | 0.8 | 4.5 (0.8) | (4.5–4.6) |
| Consumers will not have access to adequate servicing | 46.3 | 35.1 | 11.9 | 6.6 | 1.1 | 4.2 (0.9) | (4.1–4.3) |
| **Concerns about optimal benefit and adverse events** | | | | | | | |
| Level of benefit not the same as professionally fitted hearing aids | 33.2 | 40.6 | 18.6 | 6.2 | 1.4 | 3.9 (0.9) | (3.9–4.1) |
| Do not provide good value | 17.1 | 26.9 | 31.5 | 20.7 | 3.8 | 3.4 (1.1) | (3.3–3.4) |
| Consumers will give up on amplification | 14.7 | 35.7 | 23.6 | 23.4 | 2.6 | 3.4 (1.1) | (3.3–3.4) |

### 3.3. Association between HHPs' Profession and Their Primary Position with Their Views on OTC Hearing Aids

Table 3 presents the results of the Fisher's exact test showing the relationship between the HHPs' professional position and views towards OTC hearing aids. There was a significant association for the items related to *consumers having problems to identify and address common problems, lead to greater confusion, ripped off by bad actors, warranties and return period will be worse,* and *that consumers will give up on hearing aids.* Among audiologists, the odds of believing the following facts: *consumers having problems to identify and address common problems, lead to greater confusion, ripped off by bad actors, and warranties and return period will be worse* is significantly higher compared to the other HHP professions (i.e., business owners, HIS, student). Nevertheless, the odds of believing consumers will give up on amplification is significantly higher among business owners.

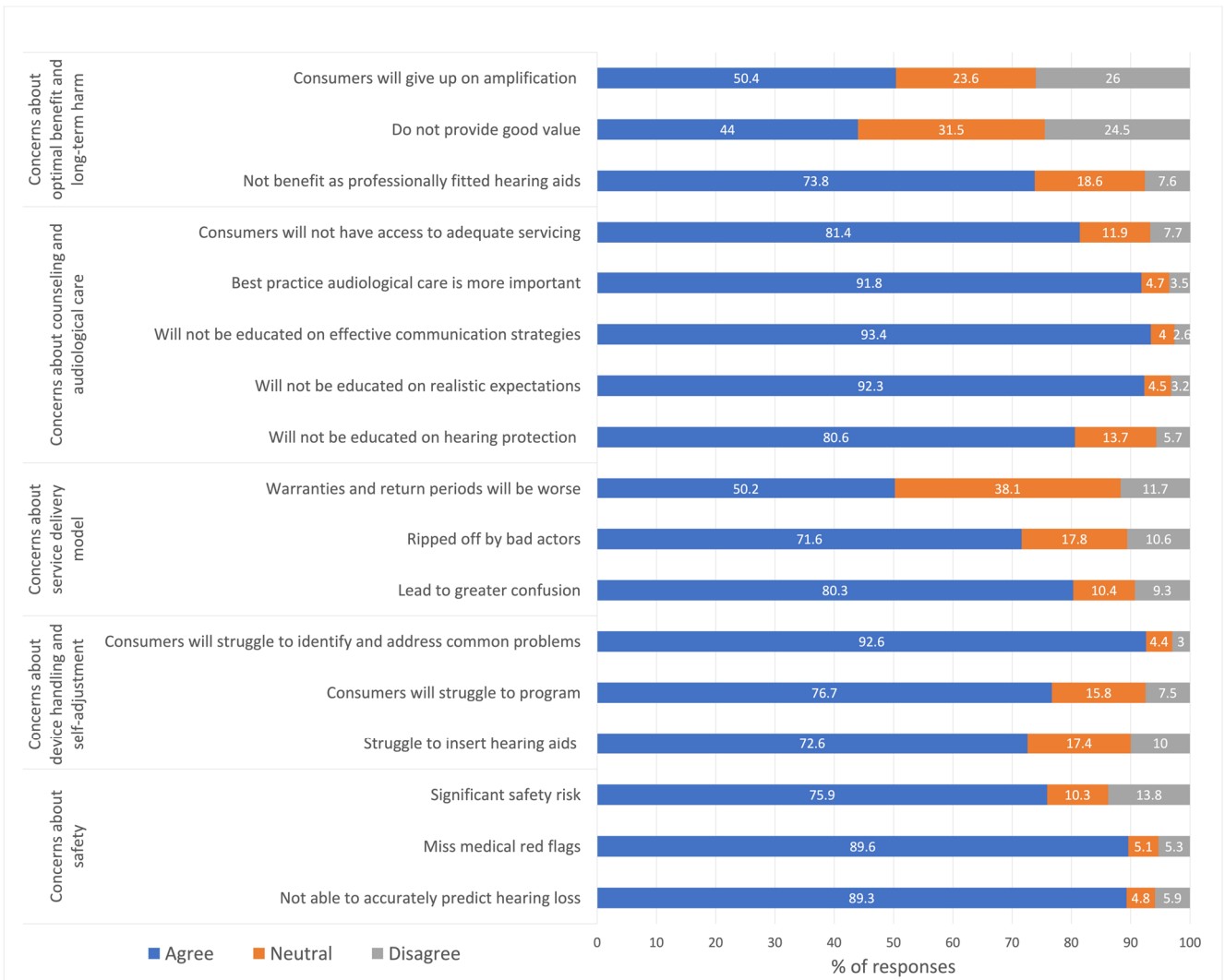

**Figure 1.** Hearing healthcare professionals' (HHPs) concerns towards over-the-counter (OTC) hearing aids.

**Table 3.** Relationship between hearing healthcare professionals' (HHPs) profession and their views on over-the-counter (OTC) hearing aids.

| | Unadjusted Odds Ratio (95% CIs) | | | | *p*-Value |
|---|---|---|---|---|---|
| | Audiologist (n = 653) | Business Owner (n = 54) | HIS (18) | Student (n = 5) | |
| **Concerns about device handling and self-adjustment** | | | | | |
| Consumers will struggle to identify and address common problems | Ref | 0.1 (0.01, 5.3) | 0.2 (0.1, 2.3) | 0.03 (0.01, 1.9) | 0.04 |
| **Concerns about service delivery model** | | | | | |
| Lead to greater confusion | Ref | 0.5 (0.1, 18.1) | 0.3 (0.1, 1.3) | 0.1 (0.01, 4.8) | 0.03 |
| Ripped off by bad actors | Ref | 0.5 (0.06, 18.2) | 0.4 (0.1, 1.5) | 0.05 (0.01, 3.7) | 0.002 |

**Table 3.** *Cont.*

| | Unadjusted Odds Ratio (95% CIs) | | | | *p*-Value |
|---|---|---|---|---|---|
| | Audiologist (n = 653) | Business Owner (n = 54) | HIS (18) | Student (n = 5) | |
| Warranties and return periods will be worse | Ref | 0.6 (0.07, 24.9) | 0.6 (0.5, 3.3) | 0.08 (0.01, 6.2) | 0.02 |
| **Concerns about optimal benefit and long-term harm** | | | | | |
| Consumers will give up on amplification | Ref | 1.1 (0.1, 45.4) | 0.3 (0.1, 1.2) | 0.00 (0.01, 8.6) | 0.01 |

Table 4 presents the results of the Fisher's exact test showing the relationship between professional position and the responses to the survey items about OTC hearing aids. There was a significant association for items related to *significant safety risk, lead to greater confusion, not benefit as professionally fit hearing aids, do not provide value,* and *that consumers will give up on hearing aids* for practice employees. In comparison with practice employees, the odds of believing *significant safety risk* is significantly higher in owner category, whereas the odds of *believing consumers will give up on amplification* is significantly higher in other positions.

**Table 4.** Relationship between hearing healthcare professionals' (HHPs) primary position and the views on over-the-counter (OTC) hearing aids.

| | Unadjusted Odds Ratio (95% CIs) | | | | *p*-Value |
|---|---|---|---|---|---|
| | Practice Employee (n = 359) | Practice Manager (n = 99) | Owner (n = 231) | Other (n = 41) | |
| **Concerns about safety** | | | | | |
| Significant safety risk | Ref | 0.47 (0.15, 2.3) | 1.2 (0.42, 4.9) | 0.5 (0.12, 4.9) | 0.007 |
| **Concerns about service delivery model** | | | | | |
| Lead to greater confusion | Ref | 0.3 (0.08, 1.9) | 0.4 (0.14, 1.5) | 0.4 (0.04, 17.2) | 0.004 |
| **Concerns about optimal benefit and long-term harm** | | | | | |
| Do not provide good value | Ref | 0.9 (0.6, 0.7) | 0.29 (0.001, 0.001) | 0.09 (0.08, 0.09) | 0.008 |
| Consumers will give up on amplification | Ref | 0.6 (0.2, 3.9) | 0.9 (0.4, 2.9) | 1.0 (0.2, 9.6) | 0.04 |

*3.4. Sub-Group of HHPs Based on Their Views of OTC Hearing Aids*

Table 5 presents cluster analysis results showing the sub-group of HHPs based on their view towards OTC hearing aids on the 17-item questionnaire. This table includes the most likely responses of participants who belong to this group on the preference questionnaire. Based on the Silhouette value, a two-cluster solution was identified. Cluster 1 included 374 (i.e., 51.2%) HHPs who mainly had concerns about all the items rated as strongly agree (14 items) or agree (3 items). We named this group 'OTC-Averse.' On the other hand, cluster two included the remaining 356 (i.e., 48.8%) had some items rated as disagree (i.e., consumers will give up on amplification) and neither agree nor disagree (i.e., do not provide good value, warranties and return periods will be worse), whereas the remaining items were rated as either agree (13 items) or strongly agree (1 item). Based on its characteristics, we named this group as 'OTC-Apprehensive'.

**Table 5.** Sub-group of hearing healthcare professionals (HHPs) based on their views on over-the-counter (OTC) hearing aids.

| Cluster 1: OTC Averse (n = 374; 51.2%) | Cluster 2: OTC Apprehensive (n = 356; 48.8%) |
|---|---|
| **Strongly agree**<br>▪ Not able to accurately predict hearing loss<br>▪ Significant safety risk<br>▪ Not benefit as professionally fitted hearing aids<br>▪ Miss medical red flags<br>▪ Consumers will struggle to program<br>▪ Lead to greater confusion<br>▪ Ripped off by bad actors<br>▪ Warranties and return periods will be worse<br>▪ Will not be educated on hearing protection<br>▪ Will not be educated on realistic expectations<br>▪ Will not be educated on effective communication strategies<br>▪ Best practice audiological care is more important<br>▪ Consumers will struggle to identify and address common problems<br>▪ Consumers will not have access to adequate servicing<br>**Agree:**<br>▪ Struggle to insert hearing aids<br>▪ Do not provide good value<br>▪ Consumers will give up on amplification | **Strongly agree:**<br>▪ Best practice audiological care is more important<br>**Agree:**<br>▪ Not able to accurately predict hearing loss<br>▪ Significant safety risk<br>▪ Not benefit as professionally fitted hearing aids<br>▪ Struggle to insert hearing aids<br>▪ Miss medical red flags<br>▪ Consumers will struggle to program,<br>▪ Lead to greater confusion<br>▪ Ripped off by bad actors<br>▪ Will not be educated on hearing protection<br>▪ Will not be educated on realistic expectations<br>▪ Will not be educated on effective communication strategies<br>▪ Consumers will struggle to identify and address common problems<br>▪ Consumers will not have access to adequate servicing<br>**Neither agree nor disagree:**<br>▪ Do not provide good value<br>▪ Warranties and return periods will be worse<br>**Disagree:**<br>▪ Consumers will give up on amplification |

## 4. Discussion

The current study examined views of HHPs regarding OTC hearing aids by asking them to rate statements relating to several potential areas of concern. In addition, the study examined the relationship between responses and key demographics (i.e., profession, primary position) as well as identifying sub-groups of HHPs based on their OTC hearing aid-rated concerns. The key results are discussed below.

Perspectives of key hearing health stakeholders such as HHPs are important for the successful implementation of new and emerging service delivery models for OTC hearing aids. In the current study, the majority of the participants expressed concerns in almost all domains examined but particularly in terms of *safety*, *counseling*, and *audiological care*. This is interesting in light of the fact that nearly half of HHPs reported that they will support patients with OTC hearing aids purchased elsewhere and a quarter reported that they will sell OTC hearing aids in their clinic or website. Many HHPs have been preparing for OTC hearing aids by moving from bundled pricing (i.e., which bundles together HHP services and the cost of the hearing aid) to unbundled pricing (which differentiates prices for services vs. hearing aids) [12]. Nevertheless, having a high level of concerns towards OTC hearing aids and service delivery models may impact how HHPs advise their patients on OTC during hearing aid consultations and/or include OTC devices in their practice. It is likely that other healthcare professionals may also have some role to play in facilitating the journey of OTC hearing aid users. A recent survey of pharmacists showed that they had a limited understanding of OTC hearing aids and were interested in increasing their knowledge on this topic [13]. Future research should examine the validity of these concerns given that research on early generation DTC hearing devices (for review see [14,15], self-fitting algorithms [16], outcomes of OTC hearing aids and service delivery models [17–21] suggest generally positive outcomes for individuals with mild-to-moderate hearing loss.

This study showed associations between demographic variables such as profession (i.e., audiologists, HIS, business owner, student) as well as primary position (i.e., practice employee, practice manager, owner, other) with some survey items. In particular, practice employees are likely to believe that *OTC hearing aids cause safety risks*, audiologists are likely

to believe that *consumers will have problems to identify and address common problems and lead to greater confusion*, and business owners are likely to believe *that the consumers will give up on amplification*. These findings are interesting and demonstrate that concerns of each sub-group of HHPs are likely relevant to the key focus of their profession and primary position. Moreover, the study also examined users based on their responses to the17-item questionnaire using cluster analysis which identified two sub-groups. The first group (i.e., OTC averse) had concerns about OTC hearing aids in all the items, whereas the second group (OTC apprehensive) had neutral or positive views on some items. However, there was little variation between these two groups further highlighting that HHPs participating in this survey tended to have significant concerns regarding OTC hearing aids. These findings need immediate attention as a recent study showed that although the audiologists in the U.S. have low burnout rates, concerns about OTC hearing aids were associated with stress ratings [22]. Efforts are needed from professional bodies to ensure adequate information and guidance are provided to audiologists about OTC hearing aids to help reduce their stress and uncertainty regarding this rapidly emerging and evolving hearing aid category.

Our study is one of the first large-scale surveys to highlight HHP concerns regarding OTC hearing aids. On the other hand, consumers may also have concerns about OTC hearing aids. A recent study examined consumer attitudes regarding the changes to U.S. Hearing Healthcare [23]. In this study, 84% of respondents expressed discomfort with the direct-to-consumer model and preferred in-person consultation with HHPs. Older adults and those who had less interest in hearing aids were less interested in pursuing OTC hearing aids. In addition, those who had experience with a previous direct-to-consumer model and those who did not have insurance coverage were more likely to pursue OTC hearing aids. Given the recent FDA ruling, the OTC category has been established and various service delivery models for dispensing OTCs are emerging, including HHP-supported OTC hearing aids. We hope that the results of our study will provide timely and crucial feedback to stakeholders including the FDA, industry, insurance payers, and professional organizations such as the American Academy of Audiology (AAA) and American Speech-Language-Hearing Association (ASHA) on how best to proceed with successful implementation of OTC hearing aids. Guidance from these organizations should include evidence on the comparative effectiveness of OTC hearing aids versus traditional prescription hearing aids, what type of users might benefit most from OTC hearing aids, what type of users are likely to need additional professional support when using hearing aids, and whether safety concerns with OTC hearing aids are valid or not. Ultimately, the various stakeholders like HHPs, regulatory agencies, payers, and industry need to work together towards their common goal of increasing the much-needed uptake of OTC hearing aids among those who will benefit from using them.

*Limitations and Further Research*

This is the first large-scale study to report the HHPs views on OTC hearing aids to the best of our knowledge. However, the study had several limitations. First, the recruitment strategy used (i.e., social media) may have resulted in sampling bias. Second, to keep the survey anonymous, some key demographic information such as age, sex, years of practice which were vital to generalizing the study results. Third, as the study used a retrospective design, there was no opportunity for researchers to contribute to the survey development and include questions that may be more neutral. Some questions were vaguely worded (e.g., *significant safety risk, lead to greater confusion*) which may have biased HHPs' responses to those items. Moreover, qualitative methodology may have been more appropriate to gather deeper insights to participants' perspectives rather than a quantitative questionnaire study. For these reasons, the results must be viewed with caution and as tentative until further studies confirm these findings. Future research should focus on examining the contextual facilitators and barriers to OTC service delivery models according to stakeholders such as users, HHPs, patient organizations, managers of the health systems,

companies manufacturing and distributing OTC HAs in different settings. Consideration of potential payers such as insurance companies is another area of future exploration.

## 5. Conclusions

OTC hearing aids were initiated as a category with the view to improving accessibility and affordability of hearing health care. However, the results of the current study suggest that HHPs have serious concerns about OTC hearing aids (e.g., *safety, handling and self-adjustment, service delivery models, counseling and audiological care, optimal benefits, and adverse events*). Moreover, the sub-group analysis highlighted that over 50% of HHPs are strongly opposed to OTC hearing aids and failed to have neutral or positive views on any item about concerns. The underlying reasons for HHP concerns and their validity should be explored by future studies. Given that OTC hearing aids are a rapidly emerging category of hearing devices, the HHP concerns cited in this study provide useful feedback regarding practical implementation solutions for stakeholders involved in improving and optimizing OTC hearing aid implementation.

**Author Contributions:** Conceptualization, V.M., A.S., A.B. and D.W.S.; methodology and formal analysis, V.M. and H.R.; writing—original draft preparation and editing, V.M.; reviewing and editing, A.S., H.R., A.B., K.C.D.S. and D.W.S. All authors have read and agreed to the published version of the manuscript.

**Funding:** V.M. and A.S. had A.B. Nexus funding from the University of Colorado to support some aspects of this study.

**Institutional Review Board Statement:** This study was conducted in accordance with Declaration of Helsinki, and approved by the institutional review board (Reference number 02606623-HUM007/0922) at the University of Pretoria, South Africa on 26 October 2022 for studies involving human subjects.

**Informed Consent Statement:** This survey was conducted by the Hearing Tracker as a part of the industry survey without the intention of research. Hence, no informed consent was obtained prior to the survey completion.

**Data Availability Statement:** The data presented in this study are available upon request from the corresponding author. The institutional policy is that a formal data-sharing agreement is required to share the data with outside researchers.

**Conflicts of Interest:** Abram Bailey is the CEO of HearingTracker.com. De Wet Swanepoel is the Co-founder and scientific advisor at the hearX Group. Karina C. De Sousa has a relationship with hearX which includes consulting. No conflict were declared from remaining authors.

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
