# Peer review of "Hearing Healthcare Professionals’ Views about Over-The-Counter (OTC) Hearing Aids: Analysis of Retrospective Survey Data"

_audiolres, doi:10.3390/audiolres13020018_

Round 1

Reviewer 1 Report

It presents interesting survey results on hearing healthcare professionals’ views about OTC hearing aids. The results could be biased as indicated in the manuscript due to several factors. However, it does show the strong concerns of HHPs. For future studies, it might be useful to collect feedback from OTC hearing aid users.

Author Response

Thank you for your feedback. We agree that the feedback from OTC hearing aid users is necessary and have made this clear in the manuscript. 

Reviewer 2 Report

Abstract

Keywords son MESH?

Introduction

It would also be interesting to include the WHO document on hearing aids, for example https://www.who.int/activities/promoting-world-hearing-forum. Or the number of people with hearing loss…

It would be interesting to address types of hearing aids, hearing aids... classification... professionals or professional teams in people with hearing disabilities (speech therapists are not mentioned, for example)...

Data Collection, understand that the questionnaire is an ad-hoc, because they have carried out an ad-hoc with several tools that assess several of the items included? Were there any sociodemographic questions? What were the inclusion and exclusion criteria? Were there people belonging to the Deaf Community who used sign language?

Likewise, a previous section of participants... I think that age, gender, years worked... are vital data for the study and do not appear

Results

What specialty were the students? I don't quite understand the primary position thing.

Table 2 is merely descriptive, I would eliminate figure 1, it is not understandable, it is very small and does not provide much information either..

I do not understand why a qualitative study was not carried out given the objective of the study, or having used the Q methodology for those premises. On the other hand in Table 5, this table is tedious, the same type of traffic light would be better understood.

Discusión

I believe that there is no contribution with the previous literature, there is no that the authors have found again with their study. On the other hand, an ad-hoc has been used, its psychometric properties have not been analyzed.

Author Response

Comment 2.1: Abstract - Keywords son MESH?

Response: Some of the key words are MeSH words (e.g., Hearing aids, Audiologists). However, we also have more specific key words to ensure they reflect the content of this manuscript.

 Comment 2.2: Introduction - It would also be interesting to include the WHO document on hearing aids, for example https://www.who.int/activities/promoting-world-hearing-forum. Or the number of people with hearing loss…

Response: We have included the reference to the World Report on hearing in the introduction as suggested.

 Comment 2.3: It would be interesting to address types of hearing aids, hearing aids... classification... professionals or professional teams in people with hearing disabilities (speech therapists are not mentioned, for example).

Response: We have provided some brief information about types of hearing aids in the second paragraph of the Introduction. However, in the USA speech therapists are not involved with hearing aids for adults. Hence, to keep the manuscript concise, we decided not to discuss other professionals who are involved with rehabilitation of hearing disabilities.

 Comment 2.4: Data Collection, understand that the questionnaire is an ad-hoc, because they have carried out an ad-hoc with several tools that assess several of the items included? Were there any sociodemographic questions? What were the inclusion and exclusion criteria? Were there people belonging to the Deaf Community who used sign language?

Response: The survey does not have any strict inclusion and exclusion criteria other than hearing healthcare professionals. We are not sure if any participants belong to Deaf community as this was not asked in the survey. In addition, limited number of sociodemographic questions were included. We have made this clear in the Methods section. Moreover, we have also highlighted these issues as study limitations.

Comment 2.5: Likewise, a previous section of participants... I think that age, gender, years worked... are vital data for the study and do not appear

Response: We fully agree with the comment. However, as the survey was not designed by the researchers (it was designed and distributed by Hearing Tracker) and due to the retrospective nature of this data, we did not have information on some of these key demographic variables. We have acknowledged this in the study limitations.

 Comment 2.6: Results - What specialty were the students? I don't quite understand the primary position thing.

Response: All students included were Doctor of Audiology (AuD) students in the USA. On the other hand, ‘primary position’ refers to whether the respondent was a business owner, manager or employee. We have clarified this in the revised manuscript.

 Comment 2.7: Table 2 is merely descriptive, I would eliminate figure 1, it is not understandable, it is very small and does not provide much information either..

Response: Thank you for your feedback. We think that the Figure 1 is necessary to provide readers a quick summary of the data. We have made some changes to Figure fonts to ensure good readability.

 Comment 2.8: I do not understand why a qualitative study was not carried out given the objective of the study, or having used the Q methodology for those premises. On the other hand in Table 5, this table is tedious, the same type of traffic light would be better understood.

Response: We agree with your comment that the qualitative study would have been appropriate to gain in-depth insights into participants views. However, as mentioned before, the study involved analysis of retrospective data obtained from the HearingTracker.com. We have clarified this in the study limitations. In addition, we have also added a statement to clarify what is included in Table 5.

Comment 2.9: Discusión - I believe that there is no contribution with the previous literature, there is no that the authors have found again with their study. On the other hand, an ad-hoc has been used, its psychometric properties have not been analyzed.

Response: We have added additional discussion with relevant literature.

Reviewer 3 Report

I enjoyed reading the paper. The red thread is clear, the writing style is adequate and the study has been well-performed.

-          What was the sampling method? Please, name it and provide the substantiation as well.

-          Why was participants’ profession not balanced in the sample?

-          I suggest extending the conclusion – please, discuss the theoretical and practical implications to a greater extent.

-          How do the authors claim not using interviews or focus groups as a research method?

-          I suggest considering the paper by Debevc et al. (2021) “Effectiveness of a self-fitting tool for user-driven fitting of hearing aids. International journal of environmental research and public health” to strengthen the manuscript.

Author Response

Comment 3.1: I enjoyed reading the paper. The red thread is clear, the writing style is adequate, and the study has been well-performed.

Response: Thank you for the positive feedback.

Comment 3.2: What was the sampling method? Please, name it and provide the substantiation as well.

Response: The study used a convenience sampling. We have clarified this in the revised manuscript.

 Comment 3.3: Why was participants’ profession not balanced in the sample?

Response: As the study was retrospective, we had no influence on data collection.

Comment 3.4: I suggest extending the conclusion – please, discuss the theoretical and practical implications to a greater extent.

Response: We have provided some additional text to highlight the practical implications of the study results. 

Comment 3.5: How do the authors claim not using interviews or focus groups as a research method?

Response: We agree that the not using a qualitative methodology would have been more appropriate for this study. We have highlighted this as a study limitation.

Comment 3.6: I suggest considering the paper by Debevc et al. (2021) “Effectiveness of a self-fitting tool for user-driven fitting of hearing aids. International journal of environmental research and public health” to strengthen the manuscript.

Response: Thank you for your suggestion. We have cited this paper in the discussion.

Reviewer 4 Report

Journal: “Audiology Research”

Type of Paper: Article

Title of the manuscript: “Hearing healthcare professionals’ views about over-the-counter 2 hearing aids”.

This article concerns a retrospective survey design study that aims to examine the concerns that hearing care professionals have with regard to certain hearing aids (OTC). By means of descriptive data analysis, fisher tests and cluster analysis, the authors derive numerous answers to the questions underlying the concern about these types of hearing aids. They conclude by noting how the concerns that emerge from the study can be used to improve these types of hearing aids.

Although the study is overall well designed and very useful for increasing knowledge about hearing devices, which are essential for the well-being of people with hearing loss, some points are unclear. Below are a few observations and questions that I hope will serve to increase the clarity of the work.

-Title: In the title, I would include a reference to the fact that the article is about a retrospective study

-Abstract: The authors should try to summarise better, especially the results section.  Furthermore, OTC acronym is included (both in the abstract and throughout the article) without defining it. Please introduce the acronym OTC both in this section and in the text.

-Introduction: Add reference line 49

-Lines 66-76 add reference for statements made

-Tables 1 and 2: insert legend specifying acronyms in table

-Table 2: Insert "HHPs" in full and the acronym in brackets to make the table easier to understand

-Discussion: try to expand the section by inserting more references to support the discussion

Author Response

Comment 4.1: This article concerns a retrospective survey design study that aims to examine the concerns that hearing care professionals have with regard to certain hearing aids (OTC). By means of descriptive data analysis, fisher tests and cluster analysis, the authors derive numerous answers to the questions underlying the concern about these types of hearing aids. They conclude by noting how the concerns that emerge from the study can be used to improve these types of hearing aids.

Although the study is overall well designed and very useful for increasing knowledge about hearing devices, which are essential for the well-being of people with hearing loss, some points are unclear. Below are a few observations and questions that I hope will serve to increase the clarity of the work.

Response: Thank your feedback. We have revised the manuscript based on your and the other reviewers comments.

Comment 4.2: Title: In the title, I would include a reference to the fact that the article is about a retrospective study

Response: We have included the reference about the study being retrospective in the title and abstract.

Comment 4.3: Abstract: The authors should try to summarise better, especially the results section.  Furthermore, OTC acronym is included (both in the abstract and throughout the article) without defining it. Please introduce the acronym OTC both in this section and in the text.

Response: We have revised the text on results in abstract and also defined the OTC acronym.

Comment 4.4: Introduction: Add reference line 49

Response: We have added a reference to this statement.

Comment 4.5: Lines 66-76 add reference for statements made

Response: We have added a reference to this statement.

Comment 4.6: Tables 1 and 2: insert legend specifying acronyms in table

Response: We have inserted acronyms to all the tables and table legends.

Comment 4.7: Table 2: Insert "HHPs" in full and the acronym in brackets to make the table easier to understand

Response: We have inserted acronyms to all the tables and table legends.

Comment 4.8: Discussion: try to expand the section by inserting more references to support the discussion

Response: We have revised the discussion section by including references to relevant previous literature.

Round 2

Reviewer 2 Report

Thanks for the changes made, I think the article has improved compared to the previous version. Just a couple of comments to modify.

Figure 1 is too small and yellow does not make it easy to read

Line 280 I understand that it is sex not gender.

Author Response

Comment: Figure 1 is too small and yellow does not make it easy to read

Response: We will request a copyeditor to ensure that the image size is appropriate for the manuscript. However, we have changed the color for Figure 1. 

Comment: Line 280 I understand that it is sex not gender.

Response: We have made this suggested change.